# Magnetic Resonance Imaging in the Clinical Care for Uveal Melanoma Patients—A Systematic Review from an Ophthalmic Perspective

**DOI:** 10.3390/cancers15112995

**Published:** 2023-05-30

**Authors:** Myriam G. Jaarsma-Coes, Lisa Klaassen, Marina Marinkovic, Gregorius P. M. Luyten, T. H. Khanh Vu, Teresa A. Ferreira, Jan-Willem M. Beenakker

**Affiliations:** 1Department of Ophthalmology, Leiden University Medical Center, 2333 ZA Leiden, The Netherlands; 2Department of Radiology, Leiden University Medical Center, 2333 ZA Leiden, The Netherlands; 3Department of Radiation Oncology, Leiden University Medical Center, 2333 ZA Leiden, The Netherlands

**Keywords:** magnetic resonance imaging, uveal melanoma, diagnosis, treatment planning, ocular oncology

## Abstract

**Simple Summary:**

In the past, eye tumours were generally not assessed with magnetic resonance imaging (MRI) due to low image quality. MRI of the eye has significantly improved in the last decade and is, therefore, used more and more to visualise the tumour and surrounding structures. This review provides an overview of how MRI can be used to improve the clinical care of patients with a uveal melanoma, the most common eye tumour in adults. MRI can help in diagnosis, because it provides anatomical and biological information which cannot be attained by the conventional ophthalmic techniques. Dimension measurements on MRI are generally in agreement with ocular ultrasound, but MRI is more reliable when the tumour is located in the anterior part of the eye. Additionally, MRI can provide information about prognosis and treatment response, without performing a biopsy.

**Abstract:**

Conversely to most tumour types, magnetic resonance imaging (MRI) was rarely used for eye tumours. As recent technical advances have increased ocular MRI’s diagnostic value, various clinical applications have been proposed. This systematic review provides an overview of the current status of MRI in the clinical care of uveal melanoma (UM) patients, the most common eye tumour in adults. In total, 158 articles were included. Two- and three-dimensional anatomical scans and functional scans, which assess the tumour micro-biology, can be obtained in routine clinical setting. The radiological characteristics of the most common intra-ocular masses have been described extensively, enabling MRI to contribute to diagnoses. Additionally, MRI’s ability to non-invasively probe the tissue’s biological properties enables early detection of therapy response and potentially differentiates between high- and low-risk UM. MRI-based tumour dimensions are generally in agreement with conventional ultrasound (median absolute difference 0.5 mm), but MRI is considered more accurate in a subgroup of anteriorly located tumours. Although multiple studies propose that MRI’s 3D tumour visualisation can improve therapy planning, an evaluation of its clinical benefit is lacking. In conclusion, MRI is a complementary imaging modality for UM of which the clinical benefit has been shown by multiple studies.

## 1. Introduction

Although magnetic resonance imaging (MRI) has been used to image uveal melanoma patients since its introduction in the 1980s [1,2], it was sparsely used in clinical practice as eye-motion resulted in a low image quality. As a result, the diagnosis and treatment of intra-ocular tumours was primarily based on ophthalmic imaging modalities, such as fundoscopy and ultrasound, conversely to tumours in other parts of the body, where advances in MR- and CT-imaging have significantly improved clinical care. In the last decade, however, technical developments have enabled the acquisition of high-quality MR-images of the eye and orbit [3,4,5,6,7,8,9,10]. As a result, MR-imaging has been proposed for different ocular conditions such as inflammation, refractive conditions, glaucoma and several malignancies [10,11,12,13,14,15,16,17,18,19,20,21].

For uveal melanoma (UM) patients specifically, multiple clinical applications of ocular MRI have been proposed [11,22,23,24,25,26,27,28,29,30]. MRI makes it possible to obtain a complete 3D visualisation of the eye and orbit, without the limited penetration depth of ultrasound or fundoscopy [31,32]. This 3D visualisation makes MRI more accurate in measuring the dimension of anterior tumours, especially if conventional ultrasound is not able to visualise the entire tumour due to its limited penetration depth or limitations in probe placement [32]. Furthermore, a study comparing MRI and histopathological findings suggests that MRI is more reliable than ultrasound in the detection of optic nerve invasion and extrascleral extension [28]. Different studies, therefore, propose methods to use MRI to further advance the treatment planning of these patients [33]. Additionally, MRI can probe different biological aspects of the tumour without the need of a biopsy. These new imaging biomarkers have proven to aid in the diagnosis and follow-up of patients with UM and other intraocular masses and may also provide prognostic information [15,26,28,34,35,36,37,38,39].

With MRI emerging as a clinically valuable imaging modality for the eye, it is important to understand its strengths and limitations from an ophthalmic perspective, especially since a high level of care has already been attained with the conventional techniques. However, as most ocular-MRI studies are described from a radiological or physics perspective, their implications for ophthalmic clinical practice are often not fully explored. Therefore, we systematically reviewed the ocular MRI literature from an ophthalmic perspective, with particular attention to the clinical implications of MRI for patients with uveal melanoma. In this review, we will describe the characteristics of UM on MRI, followed by possible clinical applications of MRI in the differential diagnosis, treatment planning and follow-up of these patients. Finally, these evaluations will be combined into indications for ocular MRI in the care of patients with an intraocular mass. 

## 2. Materials and Methods

This systematic review was carried out according to the PRISMA 2020 statement [40]. PubMed and the Cochrane Library were searched for full-text articles in English, published between January 2000 and December 2022 using the following term:
(1)(“ocular ma*” OR “intraocular ma*” “Uveal melanoma” OR “Choroidal melanoma” OR “iris melanoma”) AND (“magnetic resonance imaging [MeSH] OR “MRI”)

In Rayyan [41], all records were screened jointly by two authors (MJC, JWB) based on the title and abstract. The full text was read for all potentially eligible records. Articles were excluded if no details were provided on the use of MRI (full text only), it did not involve an intraocular mass or only considered metastatic disease outside the orbit. Additionally, studies that did not include in vivo MR-imaging were excluded. Single case studies were excluded when larger studies were available on a similar topic. These case studies are listed in Appendix A. The remaining articles were assessed to determine the contribution of MRI in the clinical care for patients with an intraocular mass. Furthermore, records were identified from reference lists of included studies. An overview of the search and inclusion process is shown in Figure 1. 

In this review, data from several studies were combined for quantitative MRI measures of the tumour perfusion and diffusivity [11,26,28,34,35,36,37,38,42,43,44,45]. Moreover, the data comparing ultrasound-based and MRI-based tumour dimensions of different studies were combined in a Bland–Altmann plot in order to provide a comprehensive evaluation of the agreement between these modalities [3,28,32,33,46,47]. The methods for this review were not previously registered.

## 3. Uveal Melanoma on MRI

Modern ocular MRI scans are preferably performed on a 3T MRI scanner with a surface coil as the increased signal to noise ratio delivers a higher diagnostic value compared to scans acquired with a head coil or a 1.5 tesla scanner [10,21,48]. These protocols preferably contain both two- and three-dimensional anatomical sequences complemented with functional imaging [10,11,21]. The two-dimensional sequences are particularly useful to evaluate anatomy, layer of origin and to assess the margins of the lesion, while the three-dimensional sequences allow for a comprehensive assessment of tumour geometry through multiplanar reconstructions. This three-dimensional evaluation proved to be particularly valuable in the determination of the tumour dimensions in the context of therapy selection, planning and follow-up [3,10,47]. In addition to these anatomical scans, functional MRI-scans provide different quantitative imaging biomarkers which can aid in the differential diagnosis, staging and assessment of therapy response [26,28,34,37,38,39]. It has been shown that these MRI scans can be performed safely and reliably at field strengths up to 7 tesla, in patients with modern intraocular lenses (IOL) or a Baerveldt glaucoma implant, after silicon oil tamponade and with tantalum markers sutured to the sclera [33,49,50,51,52,53,54,55].

### 3.1. Anatomical Evaluation

In MRI, multiple images with different contrasts are generally jointly evaluated. These different contrasts provide the complementary information needed to distinguish different pathologies such as intraocular masses, associated retinal detachment, necrosis and other treatment related effects [21,22,28,56,57,58]. For the eye, these contrasts should at least include T2-weighted scans and T1-weighted scans, the latter before and after administration of an intravenous gadolinium-based contrast agent (Figure 2A–C). 

The signal intensity characteristics of UM on these types of MR-images have been described extensively [10,11,26,28,34,36,37,45,56,59,60,61,62,63,64,65,66,67,68,69,70,71,72,73,74,75,76]. As almost all intraocular lesions, including UM, are hyperintense on T1- and hypointense on T2-weighted imaging compared to the vitreous, Ferreira [28] proposed to use the choroid and nearby extraocular muscle as reference on T1- and on T2-weighted imaging, respectively. Using these tissues as a reference, she described that UM are mostly hyperintense on T2-weighted imaging and hyperintense or isointense on T1-weighted imaging. The signal intensity on T1-weighted imaging correlated significantly to pigmentation [28,72], which is also clearly visible in Figure 3B, where the melanotic part (blue arrow) is more hyperintense than the amelanotic part (pink arrow). As opposed to retinal detachment, all UM, but also retinoblastomas and haemangiomas, enhance after contrast administration [11,28,74,77,78]. These characteristics are an important first step to differentiate UM from accompanying phenomena, such as detachments of the retina and choroid, which can be identified based on signal intensity and morphology [22,79]. The signal intensity of retinal detachment varies based on its contents [11] and can have a similar signal intensity as the UM on non-enhanced scans (Figure 2A,B, arrow). Therefore, contrast-enhanced T1-weighed scans are important to differentiate retinal detachment from UM, as the retinal detachment does not enhance (Figure 2B,C, arrow) [10,22,28]. Retinal detachment can be observed reliably on MRI [28,75]; however, small retinal detachments might be missed on MR-images with poorer image quality [48,80,81]. Haemorrhagic retinal detachment and other intraocular haemorrhages are easier to recognise in their subacute stage when they are hyperintense on native T1-weighted images and do not enhance after contrast administration [10,11]. In contrast to retinal detachment, necrosis and inflammation are better depicted on T2-weighted imaging, as is illustrated in Figure 3 [22,56]. Using the aforementioned features, MRI is able to distinguish UM from accompanying features or other pathologies, such as peripheral exudative haemorrhagic chorioretinopathy [25].

One of the advantages of MRI compared to ultrasound and fundoscopy is that the complete orbit can be evaluated. This allows for a more accurate screening for extrascleral extension, optic nerve invasion and inflammatory processes [11,15,27,28,29,30,82,83,84,85,86,87,88,89,90,91,92,93,94,95]. MRI generally outperforms ultrasound in screening for extrascleral extension (Figure 3C) [27,28,29,30]. Only small extrascleral extensions have been reported to be missed on both MRI and ultrasound [28,29]. In the screening for extrascleral extension, it is important to use fat-suppressed scans, as they provide better differentiation of a potential extrascleral extension of the tumour and orbital fat [10]. Additionally, a three-dimensional assessment of the enhancing extraocular component is advised, as the muscle insertions might be mistaken for extrascleral extension [28,29,30]. Brisse et al. reported only a limited accuracy in identifying optic nerve invasion in retinoblastoma on their contrast enhanced T1 sequences (AUC = 0.64; 95% CI, 0.55–0.72) [96]. On the other hand, MRI was found to reliably demonstrate or exclude optic nerve invasion in fourteen UM cases [28]. In this particular series, one case of optic nerve invasion was missed on ultrasound, but seen on MRI. Therefore, Seibel et al. recommend performing an MRI for all patients with a juxtapapillary located UM, as even small melanoma might invade the optic nerve [97]. 

### 3.2. Functional Scans

In addition to these anatomical assessments, functional MRI can provide multiple types of quantitative information on function and biology of an intraocular mass [15,26,28,34,35,36,37,38,39,68,72,98,99,100]. In oncology, perfusion weighted imaging (PWI) and diffusion weighted imaging (DWI) are the most commonly used functional MRI techniques and have proven to be valuable in the differential diagnosis and assessment of therapy response in other tumour sites. 

Similar to fluorescein angiography, in PWI, an image is acquired every few seconds during and after the administration of an intravenous contrast agent (Figure 4A,B). Current clinical MRI scanners can acquire a three-dimensional image of the complete eye with an isotropic resolution of at least 1.5 mm every 2 s [28,72]. As a result, the complete lesion, and not only its ventral surface, can be assessed with PWI. Additionally, the concentration of contrast agent can be determined on these images, which, through pharmacokinetic modelling [72,101], provides quantitative information of the tissue’s microvasculature [37,72,102]. 

In clinical practice, the amount of signal enhancement, the peak intensity and the temporal evolution of the enhancement are assessed [10,22,28,43,74]. This proves a relatively straightforward description of the uptake and potential outflow of contrast agent. For UM, a combined analysis of two studies showed a peak intensity of 1.60 ± 0.39 (*n* = 51) [28,43]. Absence of enhancement or lower peak intensities are indicative for retinal detachment [10,22,28] or benign lesions [43], although haemangiomas show a stronger enhancement than UM [74]. For the interpretation of these metrics, it is important to acknowledge that the lesion’s pigmentation is a confounding factor on the observed enhancement [72]. 

Temporal evolution of the contrast uptake is visualised by a Time Intensity Curve (TIC) and is commonly characterised as progressive, plateau or wash-out (Figure 4A) [28]. Similar to other masses, the TIC shape of intraocular masses is indicative of its malignancy. Benign lesions generally have a progressive or plateau TIC, whereas malignant lesions often show a wash-out or plateau TIC [34]. A combined analysis of two studies [28,34] shows that most UM showed a wash-out TIC (69%), whereas the remaining UM showed a plateau TIC (31%). As a result, a progressive curve is considered a clear indication that the lesion is of another aetiology, as it was not observed in any of the UM patients. In UM, the amount of wash-out is reported to decrease after radiotherapy and in these patients, progressive curves have been observed (Figure 4A) [46]. Interestingly, these perfusion changes were observed before changes in size became apparent, making PWI a potential early biomarker of therapy response [46]. Quantitative analysis of these data, for example, through pharmacokinetic modelling, is reported to be indicative for patient prognosis. For example, higher peak intensities and transfer rates between blood plasma and extravascular extracellular space were found in UM with monosomy 3, an important genetic marker for metastatic risk [28,37].

The second commonly used functional MRI technique in oncology is DWI, a technique which assesses the diffusion (i.e., mobility) of water molecules within a tissue. This diffusion, quantified as an apparent diffusion coefficient (ADC) [102], is a reflection of the tissue’s density and cell size and has been found to be a non-invasive biomarker of a lesion’s malignancy [15]. For the eye, obtaining DWI has been challenging due to eye motion and the magnetic field inhomogeneities present in the orbit, but several DWI techniques have been developed, enabling ocular applications of this technique [10,35,103,104,105,106].

In the orbit, a low ADC is indicative of malignant lesions, whereas benign lesions, retinal detachment, inflammation and lesions after treatment tend to have an ADC above 1.35 × 10^−3^ mm^2^/s [38,39]. Several studies measured the ADC of UM [11,28,36,37,44,45] and found a combined ADC of 1.11 ± 0.24 × 10^−3^ mm^2^/s (Figure 5). Interestingly, one of these studies consistently reported lower ADC values for both the UM and vitreous [36], illustrating the importance of locally validating the used protocol against the values presented in literature [107].

## 4. Differential Diagnosis

Conventional ophthalmic imaging, such as ultrasound and fundoscopy, is generally sufficient to differentiate UM from other intraocular masses [108,109]. However, in some patients, not all necessary criteria can be accurately evaluated due to the size and/or location of the mass or due to the presence of opaque media such as cataract, vitreous haemorrhage or massive choroidal effusion [11,28,110,111]. In these cases, MRI can contribute to the differential diagnosis, as it can assess different aspects of the lesion such as its origin, signal intensities and functional parameters. Although prospective studies on the accuracy of MR-based differential diagnosis of intraocular masses are still lacking, several studies and case reports provide indications of its current value. 

Different studies showed that by using only anatomical information, such as location, origin and signal intensity, MRI can contribute to the differential diagnosis for various intraocular masses including retinal pigment epithelium adenomas [73], cysts [112], retinoblastoma [21,77,113,114,115], Coats disease [116], uveitis [15,117], choroidal effusion [118], neurofibroma [119] and UM [120,121]. Additionally, different reports showed that anatomical MRI can identify choroidal haemangioma [74,122], schwannoma [123,124] and scleritis [125,126]. However, other studies report that relying on anatomical characteristics alone can result in an inconclusive or even erroneous interpretation of the images, as the anatomical characteristics are often similar for distinct pathologies. For example, although UM are commonly hyperintense on T1 due to their pigmentation, making this an important clue in the differential diagnosis, melanocytomas are also hyperintense on T1. Furthermore, as approximately 15% of UM are unpigmented [28,127,128], the lack of T1-hyperintensity alone is not sufficient to exclude UM from the differential diagnosis. As a result, an MR-based diagnosis without inclusion of functional scans has been reported to be inconclusive for teratocarcinosarcoma [129], malignant rhabdoid tumour [130], leiomyoma [131,132], lymphoma [133] and intraocular metastasis [75,94].

Given the different perfusion and diffusion characteristics of distinct intraocular and orbital masses, it is recommended to include functional imaging when the presence of a mass in the eye or orbit is expected [11,15,17,28,134]. Based on signal intensities alone, schwannomas, for example, can appear similar to an amelanotic UM [123], although the observed heterogeneous enhancement is less typical for UM [94,113,124]. PWI, however, provides a much stronger differentiation, as the observed progressive TIC (Figure 4) has not been found in any UM [124]. Similarly, lymphomas can be difficult to differentiate from UM on T1- and T2-weighted imaging [133]. However, the very low ADC values observed in lymphoma, 0.66 ± 0.09 × 10^−3^ mm^2^/s compared to 1.11 ± 0.24 × 10^−3^ mm^2^/s for UM, can provide a clear indication of the lesion’s diagnosis [38,135,136,137,138]. Furthermore, one study reported the use of an MRI-based radiomics model to differentiate uveal melanoma from other intraocular masses, such as ocular metastases and melanocytomas [100].

## 5. Therapy Planning

Conventionally, MRI has had a limited role in ocular radiotherapy [139,140]. A notable exception has been stereotactic radiosurgery, where MRI is reportedly being used to complement CT imaging [141,142,143,144,145,146,147] and has shown to contribute to an increase in local control [148]. However, for other forms of ocular radiotherapy, such as brachytherapy and proton beam therapy, the therapy is planned using a dome-shaped model to approximate the tumour geometry. This model is conventionally based on the tumours prominence and basal diameters as obtained by ultrasound, complemented by intra-operative data and optical imaging [139]. In contrast to this approximate description of the tumour geometry, MRI is proposed to provide a complete three-dimensional visualisation of the tumour and surrounding structures [3]. 

In general, a good agreement between ultrasound and MRI-based tumour dimensions is found (Figure 6) [30,32,33,46,47]. However, MRI was considered more reliable when the full tumour extent could not be visualised on ultrasound [32]. In these cases, a discrepancy between the modalities was often observed. Four studies [28,32,33,46] compared ultrasound- and MRI-based tumour dimension measurements using a modern MRI protocol, including three-dimensional contrast enhanced scans [48] with an isotropic acquisition resolution of at least 1.0 mm. A combined analysis of 72 patients (Figure 6) showed that, on average, the ultrasound measurements were slightly larger than the MRI measurements (*p* < 0.01, Prominence; median 6.3 mm vs. 6.1 mm and largest basal diameter (LBD); 14.7 mm vs. 14.0 mm). The full tumour extent was more often not visible in anterior tumours compared to posterior tumours (78% and 22% respectively, *p* < 0.001), which resulted in an increased median absolute difference in prominence of 0.2 mm and LBD of 1.2 mm, respectively (Figure 6C), with both MRI and ultrasound measurements performed including sclera. 

Although several studies have shown the feasibility of fully three-dimensional MRI-based therapy planning [97,135,136,149,150,151,152], none of these methods are available clinically. A recent study showed that MRI can also improve the conventional model-based, planning, as in specific cases, it provided more accurate axial length and tumour–marker distance measurements [33]. Moreover, modern planning systems such as OCTOPUS, RayOcular and Plaque Simulator do provide the option to show MR-images in the therapy planning, which can subsequently be used to verify, and, when needed, adapt, the model-based treatment plan to conform the three-dimensional visualisation of the MRI [139,153,154,155,156]. Others propose therapy planning by fusing MR-images with fundus photographs, to provide a better localisation of the fovea, one of the main organs at risk in ocular radiotherapy, which is unfortunately not visible on MRI [81]. In this context of increasing the accuracy of ocular radiotherapy, it is comforting to know that no significant changes were observed in tumour and eye shape between the prone MRI-scanning position and the sitting position used during ocular proton therapy [157]. 

Furthermore, two studies have shown the feasibility of verifying brachytherapy plaque position with 1.5 tesla MRI [152,158]. However, the clinical benefit of this application still needs to be demonstrated. 

## 6. Follow-Up

A reduction in tumour prominence or volume, as obtained by ultrasound, is clinically one of the primary signs of therapy response [46,159,160,161]. In general, a rapid reduction in tumour prominence is observed after brachytherapy, while after ocular proton beam therapy, an initial stable of even a temporarily increase in prominence is not uncommon. For UM patients, 3 months post ruthenium brachytherapy, a clinically significant reduction in prominence is already apparent on both MRI and ultrasound [46]. In proton therapy, however, this reduction can take months up to a year to occur and a temporary increase in tumour prominence is not uncommon [161]. 

Several studies showed that MRI can be used to quantify the reduction in tumour size after treatment [46,162,163,164,165]. One study prospectively compared MRI and ultrasound-based prominence measurements for patients treated with proton beam therapy or brachytherapy [46]. In this study, ultrasound and MRI were generally in agreement, but treatment-induced effects, such as extraocular inflammation, resulted in an overestimation of the tumour prominence on ultrasound in some patients at 3 and 6 months post treatment. Additionally, this study reported that the volumetric assessment provided by MRI enabled a more reliable evaluation of the small changes in volume in the first months after proton therapy.

Different studies propose that functional MR-imaging can provide early biomarkers of therapy response [26,44,46]. Favourable PWI changes, such as a decrease in wash-out, have been observed as early as 3 months after treatment, and also when changes in size were not yet apparent [46]. Other studies propose an increase in ADC as a measure for therapy response [26,44,46,163]. Although, on average, the diffusivity indeed increases after therapy, the confounding effect of (micro)necrosis might not make it a useful biomarker on the individual patient level [46,56];

Finally, exudative retinal detachment and sick tumour syndrome, a known side effect of irradiation in patients with an intraocular mass, is sometimes treated with vitrectomy with tumourectomy and silicon oil tamponade. As the silicon oil prevents accurate ultrasound imaging, follow-up with conventional ophthalmic imaging is hindered. Monitoring these patients with MRI has been reported to prevent secondary enucleation by differentiating local tumour recurrence from scarred tumour residue [53]. 

## 7. Discussion

In recent years, different MRI techniques dedicated for imaging the eye and orbit have become clinically available, enabling the acquisition of high quality MR-images of the eye in regular clinical care. As a result, an increasing number of ophthalmic applications of MRI has been proposed in the last five years, especially related to ocular oncology. This increase can also be observed in the scientific literature, where the number of ocular oncology-related MRI-studies increased from 55 in 2000–2005 to 151 for the last 5 years. Moreover, since the radiological characteristics of the most common primary ocular tumours, UM and retinoblastoma, have now been extensively described, MRI can provide a similar contribution to the care for ocular oncology patients as it does for patient groups with other tumours.

From an imaging point of view, the eye is a relatively unique organ as it can be accurately imaged with optical techniques. Conversely to other tumour sites, adequate care can often be provided for intraocular masses without radiological imaging, as the ophthalmic imaging modalities provide sufficient information for its diagnosis, treatment and follow-up [11,28]. Notwithstanding the clinical importance of these conventional modalities, multiple studies show the added value of MRI in specific cases, as MRI can provide information not attainable with optical and ultrasonic imaging. In particular, three-dimensional visualisation of the eye and tumour, and functional assessments of the lesion’s biology, especially diffusion and perfusion weighted imaging, provide clinically valuable complementary information.

Although fundoscopic and ultrasonic imaging are generally sufficient for accurate diagnosis of intraocular masses, in various situations they cannot provide a definite diagnosis, such as for lesions behind the iris, and in cases with an accompanying vitreous haemorrhage or dense cataracts. Now, the anatomical and functional characteristics of a lesion on MRI can aid in the diagnosis for these cases, instead of an invasive biopsy. For a reliable evaluation of the MR-images, it is, however, critical that the radiologist is sufficiently acquainted with the appearance of intraocular masses on MRI, especially since not all radiological characteristics of all types of intraocular masses are known, nor have a 100% specificity. As a result, definite diagnoses based on MRI alone can be challenging. However, as a comprehensive radiological description of UM is available, a differentiation between UM and other masses is, in our experience, often possible, especially in a combined evaluation with ophthalmic data. In this context, communication between the radiology and ophthalmology professions is vital. In our experience, an MRI can also be beneficial in an atypical presentation of a mass, for example, for a very young patient with a lesion which, on ophthalmic imaging, appears as a UM. While especially the functional MRI measures can provide an independent confirmation of the diagnosis, we have also seen different cases where these biomarkers combined with MRI’s visualisation of the mass’s internal structures, point towards a different diagnosis, which was then later confirmed through biopsy. In general, MRI can be used as an additional diagnostic tool, especially for atypical presentations of intraocular lesions, if visualisation is impossible with conventional ophthalmic imaging methods or in situations where the diagnosis is uncertain. 

For the treatment selection of intraocular masses, the benefit of MRI mainly depends on the type of lesion and the available treatment options. For example, in retinoblastoma, the main goal of the MRI is diagnostic confirmation and detection of local tumour extent [21]. Although for UM, three-dimensional visualisation on MRI generally provides a more accurate size determination than conventional ultrasound [32,33], the current treatment protocols effectively incorporate these uncertainties, resulting in high rates of local control (>95%) [166,167]. When, however, multiple treatment options are available for the patient based on tumour dimensions, for example, brachytherapy and proton therapy, a small change in tumour dimensions can have a direct impact on the choice of treatment modality [3]. As small UM are generally accurately assessed with ophthalmic imaging, an MRI is generally not indicated for treatment selection and planning. However, for larger and anteriorly located UM, larger differences are observed between MRI and ultrasound measurements. In these cases, especially when the full tumour extent cannot be visualised on ultrasound, we advise performing an MRI, given the >0.5 mm difference in prominence in 72% of these patients (Figure 6). In addition to these size measurements, MRI can reportedly provide a more reliable screening for optic nerve invasion and extrascleral extension. For juxtapapillary tumours, an MRI has, therefore, been advised [97]. When an extrascleral component is suspected, we do advise performing an MRI, as it might not only impact treatment selection, but also aid in surgical planning. 

Given the relatively the set dimensions of ruthenium plaques used in ocular brachytherapy, an MRI is generally of limited value for treatment planning purposes in ruthenium brachytherapy. When treating with COMS iodine plaques, which allow for a customisation of the spatial dose profile [168], MRI could be incorporated in treatment planning. Although in the Eye Physics Plaque Simulator, a planning software for these plaques, MR-images can be used as an input for the planning; the clinical benefits of this step have not yet been published. For ocular proton therapy, MRI is increasingly being used, and was considered the second most important area of future developments [169]. While the conventional model-based approach to treatment planning, as used in EyePlan and EOPP, does not rely on radiological imaging, more modern treatment planning software, including OCTOPUS and RayOcular, do provide the incorporation of MR-images in the planning. Although an evaluation of the benefit of adding three-dimensional MRI-visualisation to the ophthalmic imaging is still subject of current studies [81,135,149], we recently showed that adding MR-based measurements to the conventional model-based planning can provide a more accurate description in 20/23 patients [33].

Compared to the diagnosis and treatment of intraocular masses, the value of MRI in the follow-up after treatment has been studied less extensively. Conventionally, a reduction in tumour prominence is considered the primary sign of therapy response. UM treated with brachytherapy generally show a rapid reduction in prominence, making the use of MRI for follow-up of little clinical value for these patients. For patients treated with ocular proton therapy, however, such a reduction in prominence can take up to a year to become apparent, suggesting that an alternative biomarker of therapy response could be of value. Until present, only two studies have prospectively assessed MRI in the follow-up of these patients [26,46]. Although functional MRI, especially perfusion weighted imaging, is proposed as an early biomarker of therapy response, independent validation of these findings in a larger cohort is still needed.

With the advent of new treatments for specific types of metastatic spread of UM [170,171], MRI-based staging and prognostication of intraocular masses might become clinically relevant in the coming years. Different studies have shown that known histopathological variants, such as monosomy 3 and the presence of microvascular loops, result in different perfusion characteristics that can be detected with MRI [28,37]. Future studies should validate these findings in larger patient groups, as well as investigating the relation of perfusion-weighted MRI with other genetic factors, such as BAP1, EIF1AX and 8q status. An additional area that warrants further prospective evaluation is the benefit of MRI in the differential diagnosis. In this respect, it would be beneficial to provide a diagnostic framework that combines the information of optical and MR-imaging, as, in our experience, this combination of disciplines provides pivotal clues for an unambiguous diagnosis. 

Although performing an MRI-scan has proven to be cost-effective for specific indications, such as for treatment decision-making in tumours with an intermediate size [33,172], the wide availability and immediate access of the ophthalmic modalities will most certainly make them stay the primary imaging modality for ocular masses for the coming decade. Nevertheless, given the increase in (biological) information attainable by MRI, an increasing contribution of MRI in ocular oncology is expected.

## 8. Conclusions

With the advances of ocular MRI, a new way of visualising the eye and orbit has emerged which complements the information provided by the more commonly used fundoscopic and ultrasound imaging. MRI can aid differential diagnosis for atypical presentations of intraocular lesions, especially when visualisation is impossible or limited with conventional ophthalmic imaging methods. For these patients, the additional information provided by MRI might mitigate the need for an intra-ocular biopsy. Furthermore, when the tumour thickness and basal diameters make the patient eligible for multiple treatments, the three-dimensional visualisation of MRI can aid treatment decision making. Furthermore, we advise performing an MRI if optic nerve invasion or an extrascleral component is suspected. Finally, first evidence has been provided of the benefit of including MRI for ocular proton beam therapy planning and the use of perfusion-weighted MRI as an early biomarker for treatment response assessment.

## Figures and Tables

**Figure 1 cancers-15-02995-f001:**
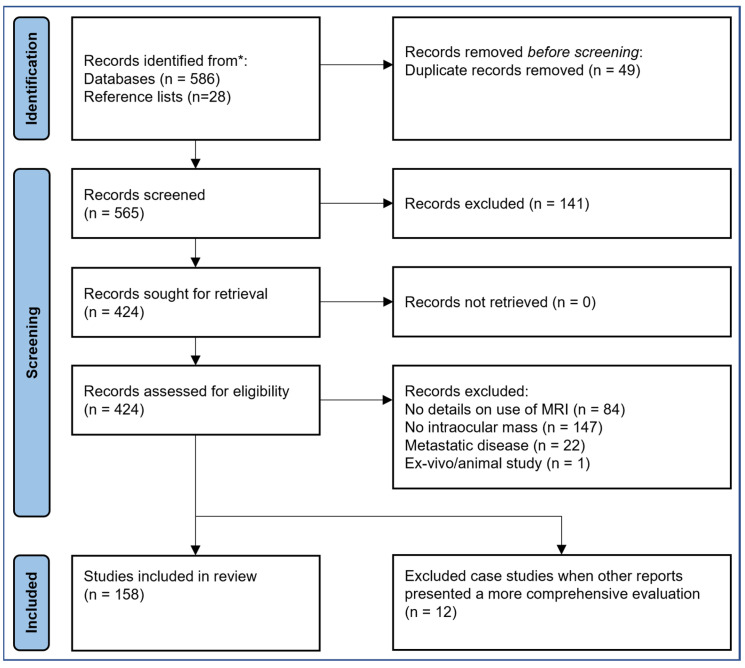
Flowchart of the search and inclusion process.

**Figure 2 cancers-15-02995-f002:**
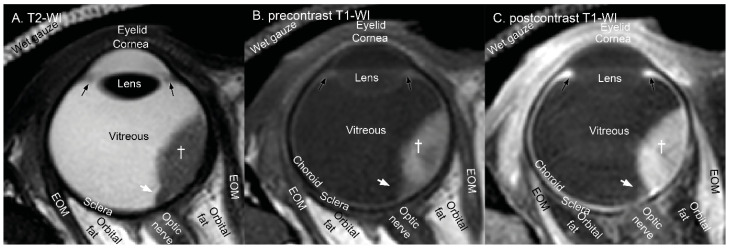
Transversal anatomical MR-images of a patient with a uveal melanoma (dagger) and associated retinal detachment (white arrow). (**A**) On T2-weighted images, most lesions are hypointense compared to the vitreous. (**B**,**C**) T1-weighted imaging before (**B**) and after (**B**) contrast showing an hyperintense mass which is enhancing. Note that the choroid, extra ocular muscles (EOM), ciliary body (black arrow) and eyelid also enhance.

**Figure 3 cancers-15-02995-f003:**
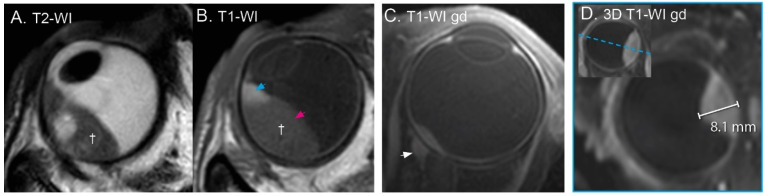
(**A**,**B**) patient with necrosis (**A**) in a bipartite uveal melanoma (cross). The amelanotic part ((**B**), pink arrow) is isointense, while the melanotic part ((**B**), blue arrow) is hyperintense compared to the choroid on T1. (**C**) Extraocular extension (white arrow) is best visualised on the contrast-enhanced T1-weighted scan with fat suppression. (**D**) A volumetric scan allows for three-dimensional visualisation of the tumour and surrounding structures and provides the most accurate dimension measurements (line) as the measurement plane (**D**) can be reconstructed perpendicular to the sclera (dotted line).

**Figure 4 cancers-15-02995-f004:**
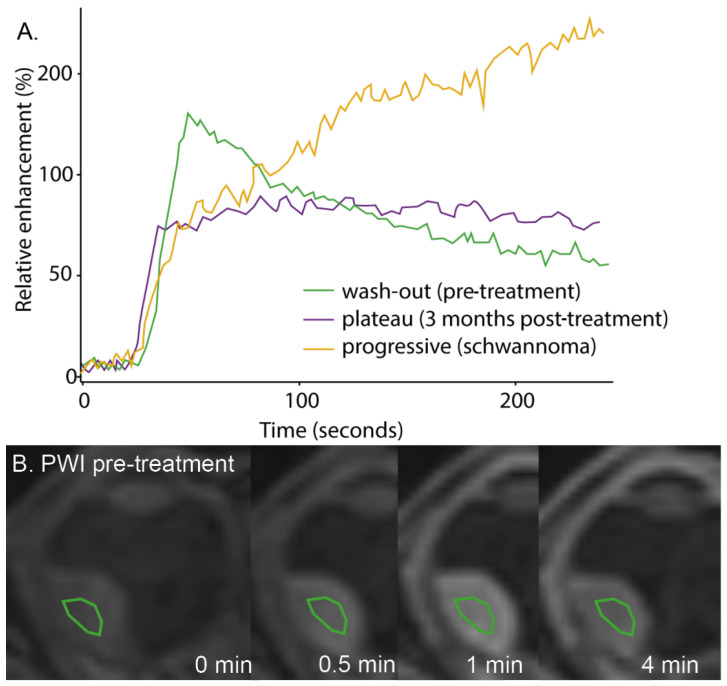
(**A**) The three different time intensity curves (TICs): A wash-out TIC of a UM patient before treatment, a plateau TIC from the same patient 3 months after treatment and a progressive curve in a patient with a schwannoma. (**B**) Four timepoints of the PWI-MRI of the same UM patient before treatment, showing an increase in signal intensity in the tumour at timepoint 0.5 and 1 min and decrease towards the end of the acquisition (4 min), resulting in the green time intensity curve (**A**).

**Figure 5 cancers-15-02995-f005:**
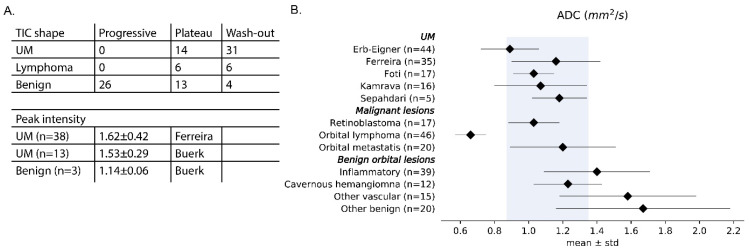
(**A**) Two-thirds of the perfusion classifications in UMs show a wash-out TIC, whereas most benign lesions have a progressive curve, which is not observed in UM. Plateau TICs are observed in both benign and malignant ocular masses [28,34,41]. On average, UMs show a relative enhancement of 1.6 after contrast administration [28,42]. (**B**) The ADC value, a measure of diffusion, of UM is 1.11 ± 0.24 × 10^−3^ mm^2^/s (gray area), which is lower than most benign orbital lesions and higher than orbital lymphoma [26,28,35,36,37,38].

**Figure 6 cancers-15-02995-f006:**
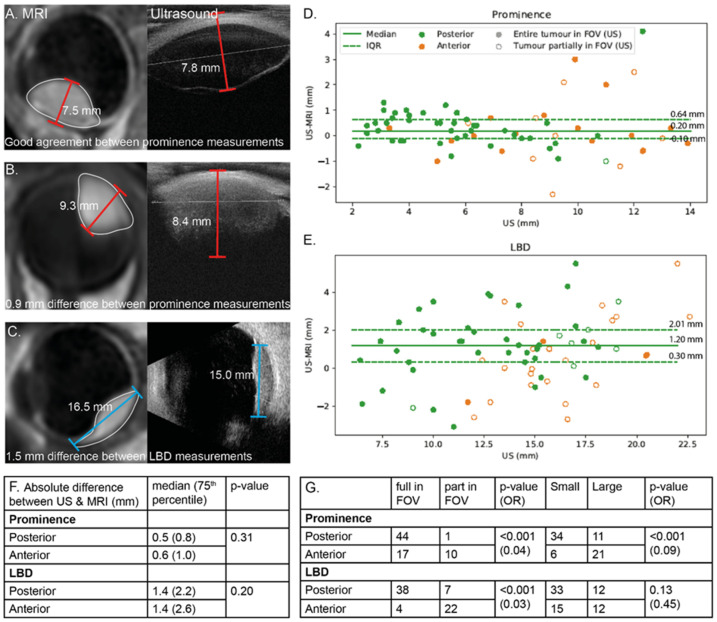
Comparison between MRI and ultrasound-based tumour measurements. Although both modalities generally are in agreement (**A**), larger differences are found for anterior tumours (**A**,**B**), orange markers in (**D**,**E**) and when the full tumour extent was not visible on ultrasound ((**C**), open circles in (**D**,**E**)). Overall, the prominence and LBD measurements were larger on ultrasound (Wilcoxon signed-rank test, *p* < 0.01) (**F**) and anterior tumours were more often not fully visible on ultrasound (*p* < 0.001) (**G**). Small tumours were defined as tumours with a prominence including sclera <7 mm and an LBD < 16 mm.

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
