# Peer review of "Magnetic Resonance Imaging in the Clinical Care for Uveal Melanoma Patients—A Systematic Review from an Ophthalmic Perspective"

_cancers, 2023, doi:10.3390/cancers15112995_

Round 1

Reviewer 1 Report

This is a review entitled “Magnetic resonance imaging in the clinical care for uveal melanoma patients – A systematic review from an ophthalmic perspective (cancers-2390448)”.

This is a well written and educative review article.

Please also add some description of scenes that MRI is the most effective method to decide such as PECHR, IOH etc.

Author Response

Reviewer #1 comments:

 This is a review entitled “Magnetic resonance imaging in the clinical care for uveal melanoma patients – A systematic review from an ophthalmic perspective (cancers-2390448)”.

This is a well written and educative review article.

Please also add some description of scenes that MRI is the most effective method to decide such as PECHR, IOH etc.

Response to reviewer #1:
We would like to thank the reviewer for their kind words and their valuable advice. The following sentences were added to address their feedback:

Line 164:              Haemorrhagic retinal detachment and other intraocular haemorrhages are easier to recognize in their subacute stage when they are hyperintense on native T1-weighted images and do not enhance after contrast administration [10, 11].

Line 168:              Using the aforementioned features, MRI is able to distinguish UM from accompanying features or  other pathologies, such as peripheral exudative haemorrhagic chorioretinopathy [25].

Reviewer 2 Report

The authors performed a systematic review about the role of magnetic resonance imaging (MRI) in eye tumors and more specifically in uveal melanoma (UM).

They found that MRI is able to detect potential therapy response and to distinguish between high- and low-risk tumors. Similar results in measuring tumor size between MRI and ultrasounds have been also described, while MRI has been supposed to have more sensitivity in measuring UMs of the anterior chamber.

The study is well designed and structured. The methods and results are also well detailed.

I think that a minor english editing of the text should be made.

I have one point:

Are there radiomics studies in which the mutational status of BAP1 or other genes commonly involved in UM have been correlated with specific MRI features? Please discuss this point by indicating it as future perspective if no studies have been published yet.

Minor english changes are advisable.

Author Response

Reviewer #2 comments:

The authors performed a systematic review about the role of magnetic resonance imaging (MRI) in eye tumors and more specifically in uveal melanoma (UM).

They found that MRI is able to detect potential therapy response and to distinguish between high- and low-risk tumors. Similar results in measuring tumor size between MRI and ultrasounds have been also described, while MRI has been supposed to have more sensitivity in measuring UMs of the anterior chamber.

The study is well designed and structured. The methods and results are also well detailed.

I think that a minor english editing of the text should be made.

I have one point:

Are there radiomics studies in which the mutational status of BAP1 or other genes commonly involved in UM have been correlated with specific MRI features? Please discuss this point by indicating it as future perspective if no studies have been published yet.

Response to reviewer #2:

We would like to thank reviewer #2 for their kind words and valuable feedback.

As opposed to the correlation between perfusion-weighted MRI findings and chromosome 3-status (addressed in line 233 and 477 of the review), no studies exist on the correlation between MRI features and BAP1-status yet. A small number of studies exist on the use of radiomics for differential diagnosis and prediction of disease-free survival in UM, however these do not include mutational status. The following sentences have been added to address this point: 

Line 299:              Furthermore, one study reported the use of an MRI-based radiomics model to differentiate uveal melanoma from other intraocular masses, such as ocular metastases and melanocytomas [101]. 

Line 479:              Future studies should validate these findings in larger patient groups, as well as investigating the relation of perfusion-weighted MRI with other genetic factors, such as BAP1, EIF1AX and 8q status.

Furthermore, several changes were made to improve the English language throughout the review, for example in lines 14, 16, 64,  128, and 425.  

Reviewer 3 Report

The authors have presented a paper about "Magnetic resonance imaging in the clinical care for uveal melanoma patients – A systematic review from an ophthalmic perspective".

The topic itself is absolutely interesting both from a clinical and a research point of view.

The paper is overall well written, and the authors have demonstrated to master the topic however I have a few points that I would like them to address in order to improve the quality if the manuscript as follows:

1) In the introduction section I would anticipate some of the concepts that emerge later in the paper such as the real contribution that MRI could provide in addition to standard US in UM such as better visualization of extrascleral extension, optic nerve invasion, anterior lesions and so on.

2) With regard to the methodology the paper is proposed as a systematic review, but I can find no reference to the PRISMA statement: please add it.

3) Why was only PubMed searched?  What about other relevant medical archives (such as Scopus and Cochrane library)? The authors should argue about their choice to limit the search only to one database: some relevant papers might have been missed with this choice

4) With regard to the follow-up after ocular radiotherapy I would suggest to provide further details about the kinetics of UM response to radiation therapy (also specifically with the use of MRI, see PMID: 35233241 for a detailed reference).

5) With regard to conclusion I believe that it could be an addition to provide the clinical indications to use MRI in UM already nowadays and not just to generally state that it will become more common in the future

Author Response

Reviewer #3 comments

The authors have presented a paper about "Magnetic resonance imaging in the clinical care for uveal melanoma patients – A systematic review from an ophthalmic perspective".

The topic itself is absolutely interesting both from a clinical and a research point of view.

The paper is overall well written, and the authors have demonstrated to master the topic however I have a few points that I would like them to address in order to improve the quality if the manuscript as follows:

We would like to thank reviewer #3 for their kind words and valuable feedback. The following changes have been made to address their feedback:

1) In the introduction section I would anticipate some of the concepts that emerge later in the paper such as the real contribution that MRI could provide in addition to standard US in UM such as better visualization of extrascleral extension, optic nerve invasion, anterior lesions and so on.

The following sentences were added to address this comment:

Line 56:                 This 3D visualisation makes MRI more accurate in measuring anterior tumours, especially if conventional ultrasound is not able to image the entire tumour due to the limited penetration depth or limitations in probe placement [32]. Furthermore, a study comparing MRI and histopathological findings suggests MRI is more reliable than ultrasound in the detection of optic nerve invasion and extrascleral extension [28].

2) With regard to the methodology the paper is proposed as a systematic review, but I can find no reference to the PRISMA statement: please add it.

We added the reference to the PRISMA statement:

Line 80:                This systematic review was carried out according to the PRISMA 2020 statement [40].

In order to be fully compliant with the PRISMA statement, the following additions were made to the text:

Line 89:                [all records were screened] jointly by two authors (MJC, JWB)
Line 102:               in a Bland-Altmann plot
Line 103:               The methods for this review were not previously registered.

3) Why was only PubMed searched?  What about other relevant medical archives (such as Scopus and Cochrane library)? The authors should argue about their choice to limit the search only to one database: some relevant papers might have been missed with this choice.

Initially, only PubMed was searched since the aim of this review was to find papers with a clinical application of MRI for uveal melanoma, and we expected to find these papers in PubMed. To be certain, we searched the Cochrane Library as well, yielding 132 records, of which 83 were not yet identified with our PubMed search. However, none of the newly found records fit the scope of this review, due to the fact that they did not describe an intraocular mass (n=69), concerned only metastatic disease (n=11), gave no details on the use of MRI (n=2) or were not a full text article (n=1). Therefore, no papers were added after this additional search. The details and results of the search have been added to the Methods section (line 81 and Figure 1).

4) With regard to the follow-up after ocular radiotherapy I would suggest to provide further details about the kinetics of UM response to radiation therapy (also specifically with the use of MRI, see PMID: 35233241 for a detailed reference).

The following sentences were added to address this comment, including the reference as suggested by the reviewer [160]:

Line 355:              A reduction in tumour prominence or volume, as obtained by ultrasound, is clinically one of the primary signs of therapy response [46, 160-162]. In general, a rapid reduction in tumour prominence is observed after brachytherapy, while after ocular proton beam therapy an initial stable of even a temporarily increase in prominence is not uncommon.

5) With regard to conclusion I believe that it could be an addition to provide the clinical indications to use MRI in UM already nowadays and not just to generally state that it will become more common in the future

To address this comment, most of the previous Conclusion section was moved to the discussion and the following clinical indications were added to the Conclusion:

Line 498:              MRI can aid differential diagnosis for atypical presentations of intraocular lesions, especially when visualisation is impossible or limited with conventional ophthalmic imaging methods. For these patients, the additional information provided by MRI might mitigate the need for an intra-ocular biopsy. Furthermore, when the tumour thickness and basal diameters make the patient eligible for multiple treatments, the three-dimensional visualisation of MRI can aid treatment decision making. We furthermore advise to perform an MRI if optic nerve invasion or an extrascleral component is suspected. Finally, first evidence has been provided of the benefit of including MRI for ocular proton beam therapy planning and the use of perfusion-weighted MRI as an early biomarker for treatment response assessment.

Round 2

Reviewer 3 Report

The authors have satisfactorily addressed all of my previous comments.